# Constrained Exploitability Descent: An Offline Reinforcement Learning Method for Finding Mixed-Strategy Nash Equilibrium

Runyu Lu [1 2]  Yuanheng Zhu [2 1]  Dongbin Zhao [2 1]

## Abstract

This paper proposes Constrained Exploitability Descent (CED), a model-free offline reinforcement learning (RL) algorithm for solving adversarial Markov games (MGs). CED combines the game-theoretical approach of Exploitability Descent (ED) with policy constraint methods from offline RL. While policy constraints can perturb the optimal pure-strategy solutions in single-agent scenarios, we find the side effect less detrimental in adversarial games, where the optimal policy can be a mixed-strategy Nash equilibrium. We theoretically prove that, under the uniform coverage assumption on the dataset, CED converges to a stationary point in deterministic two-player zero-sum Markov games. We further prove that the min-player policy at the stationary point follows the property of mixed-strategy Nash equilibrium in MGs. Compared to the model-based ED method that optimizes the max-player policy, our CED method no longer relies on a generalized gradient. Experiments in matrix games, a tree-form game, and an infinite-horizon soccer game verify that CED can find an equilibrium policy for the min-player as long as the offline dataset guarantees uniform coverage. Besides, CED achieves a significantly lower NashConv compared to an existing pessimism-based method and can gradually improve the behavior policy even under non-uniform data coverages. When combined with neural networks, CED also outperforms behavior cloning and offline self-play in a large-scale two-team robotic combat game.

---

[1]School of Artificial Intelligence, University of Chinese Academy of Sciences [2]Institute of Automation, Chinese Academy of Sciences. Correspondence to: Runyu Lu <lurunyu17@mails.ucas.ac.cn>, Yuanheng Zhu <yuanheng.zhu@ia.ac.cn>, Dongbin Zhao <dongbin.zhao@ia.ac.cn>.

*Proceedings of the 42nd International Conference on Machine Learning*, Vancouver, Canada. PMLR 267, 2025. Copyright 2025 by the author(s).

## 1. Introduction

Data-driven learning of policies is appealing, especially in scenarios where the interaction with the environment is expensive, e.g., robotic manipulation, autonomous driving, and health care. Therefore, offline reinforcement learning (RL) (Levine et al., 2020) has become an increasingly attractive research topic in recent years. However, offline RL faces an inherent challenge of distributional shift (Ross et al., 2011), which arises from visiting out-of-distribution states and actions. A direct way to address this issue is to apply policy constraints, which constrain how much the learned policy differs from the behavior policy in the dataset (Kakade & Langford, 2002; Schulman et al., 2015). In single-agent Markov decision processes (MDPs), such constraints can lead to suboptimality of the learned policy since the optimal policy is usually a pure strategy that assigns the optimal action probability one at each state (Sutton & Barto, 2018). Since the behavior policy derived from a set of offline transitions can hardly be a pure strategy, applying policy constraints with respect to the behavior policy will sacrifice the optimality of the learned policy, even if the coverage of the offline data is theoretically sufficient for learning the optimal policy (e.g., satisfying uniform concentration).

For multi-agent scenarios, the optimal solution can still be a pure strategy when it is fully cooperative. However, in adversarial games, e.g., two-player zero-sum Markov games (MGs), we usually characterize the optimal solution with the concept of Nash equilibrium (NE), which admits mixed strategies. For example, in a two-player Rock-Paper-Scissors (RPS) game, the unique NE is the mixed strategy $(\frac{1}{3}, \frac{1}{3}, \frac{1}{3})$ for both players. It is thus possible that policy constraint methods under a mixed-strategy behavior policy may not sacrifice policy optimality in MGs. While recent research in the field of game theory has developed various efficient equilibrium-learning dynamics that can be extended into model-free RL algorithms (Lanctot et al., 2017; Lockhart et al., 2019), it has not been examined if these algorithms can be further combined with existing offline learning techniques (Siegel et al., 2020; Wu et al., 2019) while preserving the guarantee to find an exact Nash equilibrium under sufficient assumptions on the data coverage.

On the other hand, while the existing pessimism-based meth-

ods are provably efficient for solving offline MDPs and MGs (see Jin et al. (2021); Xiong et al. (2023)), they have certain limitations when they are practically applied to real-world games. First, they require infinitely many samples to fully capture the stochasticity of the game and achieve the optimal solution, i.e., Nash equilibrium. However, when the game is deterministic (e.g., chess and Go), they become no longer optimal since the game transition can be determined by a finite number of samples, which are already sufficient for finding NE. Second, existing pessimism-based methods usually require information about the game horizon (see Cui & Du (2022a;b); Zhong et al. (2022); Xiong et al. (2023)) or the dynamics model (see Yan et al. (2024)) to solve Markov games. Zhang et al. (2023), as an exception, suffers from computational inefficiency. Therefore, it is still challenging to propose a practical model-free algorithm to solve infinite-horizon MGs offline with theoretical guarantees.

With the above-mentioned concerns, we ask the question:

*Is it possible to find mixed-strategy Nash equilibrium offline via model-free learning dynamics with policy constraints?*

This paper provides a positive answer to this question. Specifically, our contributions are threefold:

- We propose a novel model-free RL algorithm to find mixed-strategy Nash equilibrium in adversarial Markov games possibly with an infinite horizon. The algorithm, named Constrained Exploitability Descent (CED), is constructed by combining the policy constraint methods from offline RL with a game theoretic approach called Exploitability Descent (ED).

- We prove that, under the uniform coverage (concentration) assumption, CED converges in deterministic two-player zero-sum MGs (Theorem 5.2) without relying on a generalized gradient like ED. We further show that the min-player policy becomes unexploitable when the opponent converges to an interior point of the constrained policy space (Theorem 5.6). By exchanging the status of the two players and running CED twice, we can obtain a potential mixed-strategy NE.

- We verify the equilibrium-finding capability of CED by conducting experiments in matrix games, a tree-form game, a soccer game, and a two-team robotic combat game[1]. From a finite dataset, CED can find NE policies in all scenarios with uniform coverage, guaranteeing an ultimate NashConv significantly lower than the baseline from a pessimism-based method. As a practical offline method, CED also gradually improves the behavior policy under non-uniform data coverages. In the large-scale robotic combat game, CED clearly outperforms behavior cloning and offline self-play.

---

[1] https://github.com/lryforeal/CED-Implementation

## 2. Related Work

**Pessimism-based methods in offline games.** The recent works that directly examine offline games basically focus on sample complexity and rely on pessimistic value functions, which have been well understood in single-agent RL (Rashidinejad et al., 2021; Xie et al., 2021). These works typically append bonuses to the original Bellman operators and obtain confidence bounds on the duality gap for the policy computed from dynamic programming (Cui & Du, 2022a;b; Zhong et al., 2022; Xiong et al., 2023; Yan et al., 2024). In the theoretical analyses, corresponding concentration inequality is utilized to capture the stochasticity of the transition function. As a fundamental work, Cui & Du (2022a) proves that the coverage assumption of unilateral concentration is sufficient for finding Nash equilibrium offline in two-player zero-sum games by providing algorithms with Hoeffding/Bernstein-type bonuses. Subsequent works improve the sample complexity (see Cui & Du (2022b)) and extend the analyses to more complex scenarios concerning linear/general function approximations (see Xiong et al. (2023); Zhang et al. (2023)).

**Equilibrium-learning dynamics.** The field of algorithmic game theory (Roughgarden, 2016; Nisan et al., 2007) examines a wide range of equilibrium-learning dynamics. While the basic method of dynamic programming (or more simply, backward induction) can only deal with perfect information games like Markov games, game-theoretic learning dynamics, including Fictitious Play (FP) (Brown, 1951), Policy Space Response Oracle (PSRO) (Lanctot et al., 2017), and Exploitability Descent (ED) (Lockhart et al., 2019), can solve a broad class of games even with imperfect information. Among them, PSRO is already extended through deep reinforcement learning. ED exhibits best-iterate convergence and is conducive to offline RL extensions. While other methods like optimistic multiplicative weights update (OMWU) also enjoy last-iterate convergence (see Lee et al. (2021)), they have not been examined in infinite-horizon Markov games. Therefore, we consider ED as the basic dynamic to construct a new method for solving offline games.

## 3. Preliminaries

### 3.1. Problem Formulation

**Two-player zero-sum Markov games.** An infinite-horizon two-player zero-sum Markov game (Littman, 1994; Shapley, 1953) is represented by a tuple $\mathcal{MG} = (S, \mathcal{A}, \mathcal{B}, P, r, \gamma)$: $S$ is the state space. $\mathcal{A}$ is the action space of the max-player, who aims to maximize the cumulative reward. $\mathcal{B}$ is the action space of the adversarial min-player. $P \in [0, 1]^{|S||\mathcal{A}||\mathcal{B}| \times |S|}$ is the transition matrix. $r \in [0, 1]^{|S||\mathcal{A}||\mathcal{B}|}$ is the reward vector. $\gamma \in (0, 1)$ is the discount factor.

---

**Algorithm 1** Exploitability Descent (ED) in two-player zero-sum Markov games

---

**Input:** game model $\mathcal{MG}$ and iteration number $K$
**Output:** last iterate $\mu_K$ for max-player

**For** $k \in \{1, 2, \cdots, K\}$
  Compute $Q_k = Q^{\mu_{k-1}, \nu^\dagger}$ under $\mathcal{MG}$, where $\nu^\dagger = \text{br}(\mu_{k-1})$ is a best response against $\mu_{k-1}$
  **For** $s \in S$

  Update $\mu_k(s) = \underset{\mu(s) \in \Delta(\mathcal{A})}{\arg\min} \left\{ \sum_{a \in \mathcal{A}} \left( \mu(s, a) - \left( \mu_{k-1}(s, a) + \alpha \sum_{b \in \mathcal{B}} \nu^\dagger(s, b) Q_k(s, a, b) \right) \right)^2 \right\}$

---

In this paper, we focus on the *deterministic* two-player zero-sum MG with $P \in \{0, 1\}^{|S||\mathcal{A}||\mathcal{B}| \times |S|}$, which means that the transition is deterministic. As a multi-agent extension to the deterministic MDP (see Castro (2020)), it can describe various games ranging from real-world combat games (Chai et al., 2023) to video fighting games (Tang et al., 2023).

**Policy and value functions.** We use $(\mu, \nu)$ to represent the joint policy of the two players, where $\mu$ is the policy of the max-player (pursuers) and $\nu$ is the policy of the min-player (evader). Specifically, $\mu(s) \in \Delta(\mathcal{A})$ (resp., $\nu(s) \in \Delta(\mathcal{B})$) is the max-player's (resp., min-player's) action distribution at state $s \in S$, with $\mu(s, a)$ (resp., $\nu(s, b)$) being the probability of selecting action $a \in \mathcal{A}$ (resp., $b \in \mathcal{B}$). Furthermore, as in single-agent MDPs, define value functions $V^{\mu,\nu}(s) = \mathbb{E}\left[\sum_{t=0}^\infty \gamma^t r(s_t, a_t, b_t) | s_0 = s; \mu, \nu\right]$ and $Q^{\mu,\nu}(s, a, b) = \mathbb{E}\left[\sum_{t=0}^\infty \gamma^t r(s_t, a_t, b_t) | s_0 = s, a_0 = a, b_0 = b; \mu, \nu\right]$.

**Nash equilibrium.** A Nash equilibrium (NE) in a game corresponds to a joint policy where each individual player cannot benefit from unilaterally deviating from his/her own policy. Specifically, in a two-player zero-sum MG, an NE $(\mu^*, \nu^*)$ satisfies $V^{\mu, \nu^*} \leq V^{\mu^*, \nu^*} \leq V^{\mu^*, \nu}$ for any $\mu$ and $\nu$ at all states. An NE always exists when the game is finite, and all NEs share the same value (Shapley, 1953) $V^*(s) = V^{\mu^*, \nu^*}(s) = \max_\mu \min_\nu V^{\mu,\nu}(s) = \min_\nu \max_\mu V^{\mu,\nu}(s)$.

For fixed $\mu$ and $\nu$, define best-response value functions $V^{\mu,*}(s) = \min_\nu V^{\mu,\nu}(s)$ and $V^{*,\nu}(s) = \max_\mu V^{\mu,\nu}(s)$. Furthermore, let $\rho_0 \in \Delta(S)$ be an initial state distribution and define $\text{NashConv}(\mu, \nu) = \mathbb{E}_{s \sim \rho_0} [V^{*,\nu}(s) - V^{\mu,*}(s)]$. NashConv equals to the sum of the *exploitability* of the each player's policy and corresponds to the *duality gap* from the minimax perspective in two-player zero-sum games. For an arbitrary NE $(\mu^*, \nu^*)$, we have $\text{NashConv}(\mu^*, \nu^*) = 0$.

In this paper, we aim to find approximate Nash equilibria, which are the joint policies with NashConv close to zero. An important property of NE in two-player zero-sum games is that if $(\mu_1, \nu_1)$ and $(\mu_2, \nu_2)$ are both NEs, then $(\mu_1, \nu_2)$ and $(\mu_2, \nu_1)$ are also NEs. Therefore, it is reasonable to unilaterally learn the equilibrium policy for the max-player and the min-player. An NE policy can be constructed by combining any two of the individual equilibrium policies.

### 3.2. Exploitability Descent

Exploitability Descent (ED) (Lockhart et al., 2019) is a game-theoretic approach that generalizes the classic convex-concave optimization for solving matrix games. The core idea is to iteratively update the current policy along the gradient computed against a best response from the opponent. Compared to the methods of fictitious play (Brown, 1951) and regret minimization (Hart & Mas-Colell, 2000), ED exhibits *best-iterate convergence* rather than *average-iterate convergence* in two-player zero-sum games. Therefore, ED can be readily extended to online RL algorithms (like Zhu & Zhao (2022)) with policies parameterized by neural networks. In two-player zero-sum Markov games, ED for optimizing max-player policy $\mu$ is shown in Algorithm 1.

Define the utility function $u(\mu, \nu) = \mathbb{E}_{s_0 \sim \rho_0}[V^{\mu,\nu}(s_0)]$. For each $(s, a)$, $\sum_{b \in \mathcal{B}} \nu^\dagger(s, b) Q_k(s, a, b)$ can make up a generalized gradient of $\mu_{k-1}$'s worst-case utility $\nabla_{\mu(s,a)} u(\mu, \text{br}(\mu)) \in \partial \min_\nu u(\mu, \nu)$ (Clarke, 1975). Following the generalized gradient, $\mu_k$ can approach a local optimum $\hat{\mu}$ of the minimax problem $\max_\mu \min_\nu u(\mu, \nu)$. To optimize min-player policy $\nu$, we run Algorithm 1 again by exchanging the status of $\mu$ and $\nu$ and using an opposite reward. Then, $(\hat{\mu}, \hat{\nu})$ constructs a potential Nash equilibrium.

### 3.3. Policy Constraint Methods

In offline RL, the training process is always affected by action distributional shift (Kumar et al., 2019), which is one of the largest obstacles for model-free applications of the learning dynamics like Algorithm 1. In single-agent scenarios, the effect can be weakened by applying constraints to the learned policy $\pi$ to keep it close to the behavior policy $\pi_\beta$, which corresponds to the state-action distribution of the offline data. This ensures that the process of value estimation has a lower chance of considering the out-of-distribution actions. The extrapolation error in value estimation can be thus mitigated at the expense of policy suboptimality.

Such constraints are commonly realized using direct *policy constraints* on the policy update (Siegel et al., 2020) or indirect *policy penalties* on the value functions (Wu et al., 2019). Both methods require using a certain measure $D(\cdot, \cdot)$ (e.g.,

---

**Algorithm 2** Constrained Exploitability Descent (CED)

---

**Input:** offline dataset $\mathcal{D}$, discount factor $\gamma$, and iteration number $K$
**Output:** last iterate $\nu_K$ for min-player

Set policy constraint measure $D(\cdot, \cdot)$ and range $\delta$, policy penalty parameter $\epsilon$, and learning rate $\alpha$
Extract non-repetitive transition set $\mathcal{D}^*$, state set $S$, and action sets $\mathcal{A}, \mathcal{B}$ from $\mathcal{D}$
Compute state distribution $\rho_{\mathcal{D}}$ and behavior policy $(\mu_\beta, \nu_\beta)$ from $\mathcal{D}$
{% *Evaluate the value function under behavior policy*}

Compute $Q^{\mu_\beta, \nu_\beta} = \underset{Q}{\arg\min} \left\{ \sum_{(s,a,b,r,s') \in \mathcal{D}^*} \left( Q(s,a,b) - \left( r(s,a,b) + \gamma \mathbb{E}_{\substack{a' \sim \mu_\beta(s') \\ b' \sim \nu_\beta(s')}} \left[ Q(s',a',b') \right] \right) \right)^2 \right\}$

Initialize $Q_0 = Q^{\mu_\beta, \nu_\beta}, \mu_0 = \mu_\beta, \nu_0 = \nu_\beta$

**For** $k \in \{1, 2, \cdots, K\}$
    {% *Apply Bellman operator to the current value function*}

    Update $Q_k = \underset{Q}{\arg\min} \left\{ \sum_{(s,a,b,r,s') \in \mathcal{D}^*} \left( Q(s,a,b) - \left( r(s,a,b) + \gamma \mathbb{E}_{\substack{a' \sim \mu_{k-1}(s') \\ b' \sim \nu_{k-1}(s')}} \left[ Q_{k-1}(s',a',b') \right] \right) \right)^2 \right\}$

    **For** $s \in S$
        {% *Update $\mu$ along ED-like gradient under policy constraint*}

        Update $\mu_k(s) = \underset{\substack{\mu(s) \in \Delta(\mathcal{A}), \text{ s.t.} \\ D(\mu(s), \mu_\beta(s)) \leq \delta}}{\arg\min} \left\{ \sum_{a \in \mathcal{A}} \left( \mu(s,a) - \left( \mu_{k-1}(s,a) + \alpha \rho_{\mathcal{D}}(s) \sum_{b \in \mathcal{B}} \nu_{k-1}(s,b) Q_k(s,a,b) \right) \right)^2 \right\}$

    **For** $s \in S$
        {% *Compute approximate best response $\nu$ under policy penalty*}

        Compute $\nu_k(s) = \underset{\nu(s) \in \Delta(\mathcal{B})}{\arg\max} \left\{ \sum_{b \in \mathcal{B}} \nu(s,b) \left( -\sum_{a \in \mathcal{A}} \mu_k(s,a) Q^{\mu_\beta, \nu_\beta}(s,a,b) \right) - \epsilon D_{\text{KL}} \left( \nu(s), \nu_\beta(s) \right) \right\}$

---

KL-divergence) to describe the closeness of two policies.

The following policy update formula is an example of applying direct policy constraints:

$$\pi_k(s) = \underset{\substack{\pi(s) \in \Delta(\mathcal{A}), \text{ s.t.} \\ D(\pi(s), \pi_\beta(s)) \leq \delta}}{\arg\max} \left\{ \mathbb{E}_{a \sim \pi(s)} \left[ Q^{\pi_k}(s,a) \right] \right\}$$

In comparison, a regularized value is computed when using indirect policy penalties:

$$\pi_k(s) = \underset{\pi(s) \in \Delta(\mathcal{A})}{\arg\max} \left\{ \mathbb{E}_{a \sim \pi(s)} \left[ Q(s,a) \right] - \epsilon D(\pi(s), \pi_\beta(s)) \right\}$$

For direct policy constraints, the optimality of the learned policy is preserved only when the behavior policy $\pi_\beta$ is close enough to the true optimal policy, which is in theory a pure strategy in single-agent scenarios. However, this is unlikely to happen since $\pi_\beta$ is derived from an offline dataset. For indirect policy penalties, they face the same problem since the resulting solution could never be a pure strategy (see Lemma 4.1 for the case of KL-divergence).

## 4. Constrained Exploitability Descent

For adversarial games, even if we only apply a constraint to the computation of the best response $\nu^\dagger$ for the min-player

in Algorithm 1, the resulting max-player policy $\mu$ will surely deviate from the equilibrium of the original game for the same reason in single-agent scenarios. Surprisingly, we find that it is possible to instead keep the min-player policy $\nu$ unexploitable. We will further explain it through our subsequent mathematical derivations in theoretical analysis. With this observation, we propose an offline equilibrium-learning algorithm under policy constraints and call it Constrained Exploitability Descent (CED; see Algorithm 2).

CED inherits the basic structure of ED in each iteration. A $Q$ value is computed, the current $\mu$ is updated, and a best response $\nu$ is computed in preparation for the next iteration. However, CED has multiple differences in detail:

- $Q_k$ is updated from the last $Q_{k-1}$ rather than directly solved under the current Bellman equation.

- The update of $\mu$ at each state $s \in S$ is under a direct policy constraint $D(\mu(s), \mu_\beta(s)) \leq \delta$. An additional factor $\rho_{\mathcal{D}}(s)$ is also appended after the learning rate $\alpha$.

- The computation of $\nu$ is based on $Q^{\mu_\beta, \nu_\beta}$ (without estimating $Q^{\mu_k, \nu_k}$) and is under a KL-divergence penalty $D_{\text{KL}}(\nu(s), \nu_\beta(s))$ with a regularization parameter $\epsilon$.

Note that $\nu_k$ can still be viewed as an approximate best response to the current $\mu_k$ when $(\mu_k, \nu_k)$ is kept close to $(\mu_\beta, \nu_\beta)$. As a result, the best iterate of $\mu$ locally minimizes exploitability in a regularized game. However, under the additional KL-divergence regularization, now $\nu_k$ has a unique solution with a closed-form expression (see Lemma 4.1), which allows $\mu$ to update along a deterministic gradient rather than an arbitrary generalized gradient. This mitigates the problem that following a generalized gradient can lead to recurrence around a local optimum, making it possible to prove last-iterate (rather than best-iterate) convergence.

**Lemma 4.1** (Uniqueness of $\nu$ in CED). *In Algorithm 2, $\nu_k(s, b)$ can be uniquely determined by computing:*

$$\frac{\nu_\beta(s, b) \exp\left(-\frac{1}{\epsilon} \sum\limits_{a \in \mathcal{A}} \mu_k(s, a) Q^{\mu_\beta, \nu_\beta}(s, a, b)\right)}{\sum\limits_{b' \in \mathcal{B}} \nu_\beta(s, b') \exp\left(-\frac{1}{\epsilon} \sum\limits_{a \in \mathcal{A}} \mu_k(s, a) Q^{\mu_\beta, \nu_\beta}(s, a, b')\right)}$$

When $\epsilon > 0$, the min-player policy $\nu$ is a mixed strategy and no longer an exact best response to the max-player policy $\mu$. As a result, the limit point of $\mu$ deviates from the solution to the original minimax problem. Instead, we will prove in the following section that $\nu_k$ approaches an unexploitable min-player policy $\hat{\nu}$. By exchanging the status of max-player and min-player in the game and running Algorithm 2 again, we can also obtain an unexploitable max-player policy $\hat{\mu}$. The joint policy $(\hat{\mu}, \hat{\nu})$ constructs a potential Nash equilibrium.

## 5. Theoretical Analysis

In this section, we theoretically show that it is possible for CED (Algorithm 2) to find an exact Nash equilibrium with the following two steps: First, we prove that CED can converge to a stationary point $(\bar{Q}, \bar{\mu}, \bar{\nu})$ (Section 5.1). Second, we prove that the min-player policy $\bar{\nu}$ at the stationary point of CED is unexploitable, like any mixed-strategy Nash equilibrium of full support (Section 5.2). All of the omitted proofs are provided in Appendix A.

Throughout our analysis, we require the *uniform coverage* assumption, which means that the non-repetitive transition set $\mathcal{D}^*$ derived from the dataset $\mathcal{D}$ covers all state-action tuples $(s, a, b)$. In Cui & Du (2022a), this assumption is called uniform concentration, and a weaker assumption named unilateral concentration is analyzed. By constructing a counterexample where the exact NE becomes impossible to learn, they proved that unilateral concentration is somewhat necessary for finding Nash equilibrium offline. However, when the NE is a completely mixed strategy (e.g., the unique NEs of the matrix games in Section 6.1), unilateral concentration is equivalent to uniform concentration. Therefore, the uniform coverage assumption can be necessary for our theoretical analysis on finding mixed-strategy Nash equilibrium.

### 5.1. Convergence of CED

Lemma 5.1 shows the explicit gradient of utility function $u(\mu, \nu) = \mathbb{E}_{s \sim \rho_0}[V^{\mu, \nu}(s)]$ with respect to $\mu$. This can be viewed as an application of the policy gradient theorem in MDPs (Sutton et al., 1999) to multi-agent scenarios.

**Lemma 5.1** (Policy Gradient in MG). *Let $\rho^{\mu, \nu}(s) = \sum_{k \geq 0} \gamma^k \Pr(s|k; \mu, \nu)$, where $\Pr(s|k; \mu, \nu)$ is the probability of reaching state $s$ at time step $k$ under joint policy $(\mu, \nu)$. Then, it holds for all $s \in S$ and $a \in \mathcal{A}$:*

$$\frac{\partial u(\mu, \nu)}{\partial \mu(s, a)} = \rho^{\mu, \nu}(s) \sum_{b \in \mathcal{B}} \nu(s, b) Q^{\mu, \nu}(s, a, b)$$

Using Lemma 4.1 and Lemma 5.1, we are able to demonstrate the convergence of CED (Theorem 5.2) under an approximation about the state visitation probability $\rho$.

**Theorem 5.2** (Convergence of CED). *When $\rho^{\mu, \nu}$ approximates the true state distribution $\rho_\mathcal{D}$ of the dataset $\mathcal{D}$, CED with sufficiently small $\alpha$ and $\frac{1}{\epsilon}$ will converge to a stationary point $(\bar{Q}, \bar{\mu}, \bar{\nu})$ under the uniform coverage assumption.*

*Proof.* By Lemma 4.1, $\nu_k$ is uniquely determined by $\mu_k$. As $\mathcal{D}^*$ covers all $(s, a, b)$ tuples and the MG is deterministic, $Q_{k+1}$ in CED approximates the true value $Q^{\mu_k, \nu_k}$ when $\mu$'s learning rate $\alpha$ is close to zero. Therefore, we only need to consider the convergence of $\mu$. By Lemma 5.1, we have:

$$\frac{\partial u(\mu_k, \nu(\mu_k))}{\partial \mu_k(s, a)} = \frac{\partial u(\mu_k, \nu_k)}{\partial \mu_k(s, a)} + \sum_{b \in \mathcal{B}} \frac{\partial u(\mu_k, \nu_k)}{\partial \nu_k(s, b)} \frac{\partial \nu_k(s, b)}{\partial \mu_k(s, a)}$$

$$= \sum_{b \in \mathcal{B}} \left( \begin{array}{l} \rho^{\mu_k, \nu_k}(s) \nu_k(s, b) Q^{\mu_k, \nu_k}(s, a, b) \\ + \frac{\partial u(\mu_k, \nu_k)}{\partial \nu_k(s, b)} \frac{\partial \nu_k(s, b)}{\partial \mu_k(s, a)} \end{array} \right)$$

Note that $\frac{\partial \nu_k(s, b)}{\partial \mu_k(s, a)} \to 0$ when $\frac{1}{\epsilon} \to 0$ (see Appendix A.3 for details). When $\rho^{\mu, \nu}$ approximates $\rho_\mathcal{D}$, we have $\frac{\partial u(\mu_k, \nu_k)}{\partial \mu_k(s, a)} = \rho_\mathcal{D}(s) \sum_{b \in \mathcal{B}} \nu_k(s, b) Q_{k+1}(s, a, b)$. Therefore, $\mu_k$ in CED updates along the gradient of $u(\mu, \nu(\mu))$ at a sufficiently small learning rate $\alpha$. As a result, $\mu$ will converge to a local maximum $\bar{\mu}$ for $u(\mu, \nu(\mu))$, which implies CED will converge to a stationary point $(\bar{Q}, \bar{\mu}, \bar{\nu})$. $\square$

Theorem 5.2 provides a direct convergence guarantee for CED without relying on a generalized gradient like ED. Besides, compared to ED's underlying assumption that $\rho^{\mu, \nu}$ is uniform, the assumption of $\rho^{\mu, \nu} \approx \rho_\mathcal{D}$ is more realistic. The policy constraints employed in CED will keep $(\mu, \nu)$ close to the behavior policy $(\mu_\beta, \nu_\beta)$ derived from $\mathcal{D}$.

### 5.2. Relationship to Nash Equilibrium

Now we further show that the min-player policy $\bar{\nu}$ at the stationary point of CED follows a property of mixed-strategy Nash equilibrium in MGs, namely, being unexploitable.

**Definition 5.3** (Unexploitable). We say a joint policy $(\mu, \nu)$ in an MG is unexploitable if both $\mu$ and $\nu$ are unexploitable with respect to each other. Specifically, $\forall s \in S$:

$\mu$ is unexploitable: $\sum_{a \in \mathcal{A}} \mu(s, a) Q^{\mu, \nu}(s, a, b) = c_\mu^\nu(s), \forall b$

$\nu$ is unexploitable: $\sum_{b \in \mathcal{B}} \nu(s, b) Q^{\mu, \nu}(s, a, b) = c_\nu^\mu(s), \forall a$

Intuitively, a policy $\mu$ is unexploitable with respect to an opponent policy $\nu$ in an MG if all of the opponent actions $b$ have the same constant value $c_\mu^\nu(s)$ under each $s \in S$. As a result, the opponent cannot exploit $\mu$ by deviating from $\nu$ at any state. Lemma 5.4 shows this property can characterize the mixed-strategy Nash equilibria with full support.

**Lemma 5.4** (Property of Interior NE). *If a Nash equilibrium $(\mu^*, \nu^*)$ in an MG has full support on the action space (thus being an interior point of the joint policy space), then $(\mu^*, \nu^*)$ is unexploitable.*

Now we demonstrate that $\bar{\nu}$ at any stationary point of CED is also an unexploitable min-player policy in the MG. We first provide an auxiliary lemma to show the update of $\mu$ at each state $s \in S$ can be equivalently enforced within the hyperplane of probability simplex, where $\sum_a \mu(s, a) = 1$.

**Lemma 5.5** (Update Projection). *Let $z_a^s$ be the original update $\alpha \rho_{\mathcal{D}}(s) \sum_{b \in \mathcal{B}} \nu_k(s, b) Q_{k+1}(s, a, b)$ for $\mu_k(s, a)$ in CED. Let $y = \sum_{a \in \mathcal{A}} z_a^s$ be the summation over $\mathcal{A}$ and define the projected update as $p_a^s = z_a^s - \frac{y}{|\mathcal{A}|}$. Then, replacing all $z_a^s$ with $p_a^s$ results in the same $\mu_{k+1}(s)$ in CED.*

We call $p_a^s$ a projected update because $\sum_{a \in \mathcal{A}} p_a^s = 0$ and $(\mu(s, a) + p_a^s)_{a \in \mathcal{A}}$ is kept in the hyperplane of probability simplex. Using Lemma 5.5, we can prove that $\bar{\nu}$ is unexploitable under an interior point assumption (sufficient also for the theoretical analysis of ED (Lockhart et al., 2019)).

**Theorem 5.6** (Unilateral Unexploitability). *Let $\Pi(s) = \Pi_1(s) \cap \Pi_2(s)$ be the feasible region for $\mu(s)$, where $\Pi_1(s)$ is the probability simplex and $\Pi_2(s)$ is the region induced by the constraint $D(\mu(s), \mu_\beta(s)) \leq \delta$. For any stationary point $(\bar{Q}, \bar{\mu}, \bar{\nu})$ of CED, if $\bar{\mu}(s)$ is an interior point of $\Pi(s)$ for all $s \in S$, then $\bar{\nu}$ is an unexploitable policy with respect to $\bar{\mu}$ under the uniform coverage assumption.*

*Proof.* As $\mathcal{D}^*$ covers all $(s, a, b)$ tuples and the MG is deterministic, a stable $\bar{Q}$ with respect to $(\bar{\mu}, \bar{\nu})$ in CED corresponds to the true value $Q^{\bar{\mu}, \bar{\nu}}$. Since $(\bar{\mu}(s, a) + p_a^s)_{a \in \mathcal{A}}$ is in the hyperplane of $\Pi(s)$ and $\bar{\mu}$ is stable with respect to $(\bar{Q}, \bar{\nu})$, we can consider the following two cases:

Case i: $(\bar{\mu}(s, a) + p_a^s)_{a \in \mathcal{A}}$ belongs to $\Pi(s)$. Then, $\bar{\mu}(s) = (\bar{\mu}(s, a) + p_a^s)_{a \in \mathcal{A}} \Rightarrow p_a^s = 0, \forall a \in \mathcal{A}$.

Case ii: $(\bar{\mu}(s, a) + p_a^s)_{a \in \mathcal{A}}$ does not belong to $\Pi(s)$. Then, $\bar{\mu}(s)$ is the closest point in $\Pi(s)$ with respect to the point $(\bar{\mu}(s, a) + p_a^s)_{a \in \mathcal{A}}$ in the same hyperplane. This contradicts the assumption that $\bar{\mu}(s)$ is an interior point of $\Pi(s)$.

Therefore, it holds for all $s \in S$ that $p_a^s = 0, \forall a \in \mathcal{A}$, which further implies that $z_a^s = c(s), \forall a \in \mathcal{A}$. As a result, $\sum_{b \in \mathcal{B}} \bar{\nu}(s, b) \bar{Q}(s, a, b) = \sum_{b \in \mathcal{B}} \bar{\nu}(s, b) Q^{\bar{\mu}, \bar{\nu}}(s, a, b) = c(s), \forall a \in \mathcal{A}$, which means that the min-player policy $\bar{\nu}$ is unexploitable with respect to $\bar{\mu}$. □

With Theorem 5.6, if we run Algorithm 2 twice by exchanging the status of the two players and both max-player policies converge to an interior point, then the last iterates $(\mu, \hat{\nu})$ and $(\hat{\mu}, \nu)$ can construct an unexploitable joint policy $(\hat{\mu}, \hat{\nu})$. Policy constraints play an important role in supporting this claim. On the one hand, the distance between $\mu$ and $\mu_\beta$ is restricted by the direct policy constraint. On the other hand, the indirect policy penalty can also bound the distance between $\hat{\mu}$ and $\mu_\beta$ (corresponding to the $\nu_k$ and $\nu_\beta$ in Algorithm 2 after the status exchange; see Lemma A.1 in Appendix A.6 for an explicit bound). Since both $\mu$ and $\hat{\mu}$ are close to $\mu_\beta$ under policy constraints, we have $Q^{\mu, \hat{\nu}} \approx Q^{\mu_\beta, \hat{\nu}} \approx Q^{\hat{\mu}, \hat{\nu}}$, which implies that $\hat{\nu}$ is also unexploitable with respect to $\hat{\mu}$. By symmetry, it is direct to show that the joint policy $(\hat{\mu}, \hat{\nu})$ is unexploitable and thus constructs a potential mixed-strategy Nash equilibrium.

In Appendix C, we combine the existing theory to provide an overall explanation about the advantages of the CED method. In the next section, we will further verify through experiments that CED can practically find NE policies under uniform coverage. Under non-uniform data coverages, we also find that CED can gradually improve the behavior policy and eventually obtain a competitive policy.

## 6. Experiments

We conduct experiments for tabular CED in matrix games, a tree-form game, and a soccer game. We also evaluate CED under function approximations in a robotic combat game.

### 6.1. Matrix Game

We first examine if CED manages to find mixed-strategy Nash equilibrium in static matrix games. We consider two games with two valid actions from $\{1, 2\}$ for both players. The payoff matrices are $\mathcal{M}_1 = ((1, 0), (-2, 4))$ and $\mathcal{M}_2 = ((1, 0), (-2, 3))$, where the rows correspond to the actions of the max-player and the columns correspond to the actions of the min-player. $\left(\mu^*(1) = \frac{6}{7}, \nu^*(1) = \frac{4}{7}\right)$ and $\left(\mu^*(1) = \frac{5}{6}, \nu^*(1) = \frac{1}{2}\right)$ are the unique NEs in the games.

The learning curves of $(\mu, \nu)$ for CED ($\alpha = 0.01, \epsilon = 1.0$) under uniform coverage $\left(\mu_\beta(1) = \frac{1}{2}, \nu_\beta(1) = \frac{1}{2}\right)$ are shown in Figure 1. The y-axis indicates the probability of choosing action 1 under the corresponding policy. The dashed lines show the action probabilities of the unique NE policy. In both games, CED manages to learn the equilibrium policy $\nu = \nu^*$ for the min-player. This result is

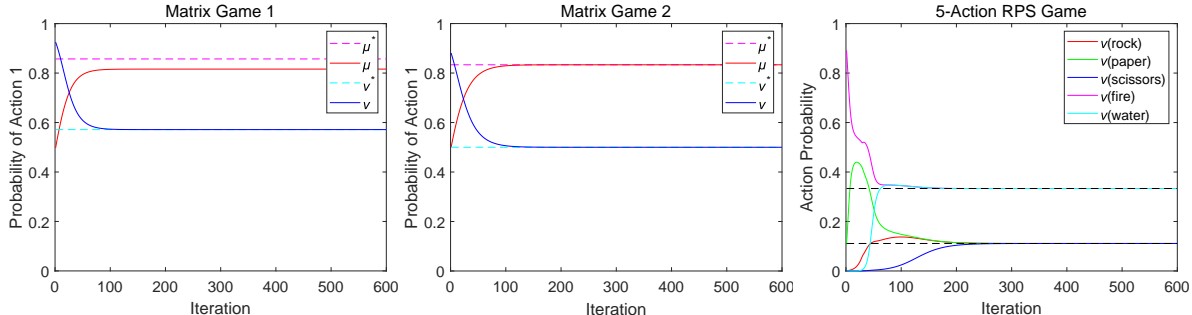

*Figure 1.* CED learning curves in the matrix games

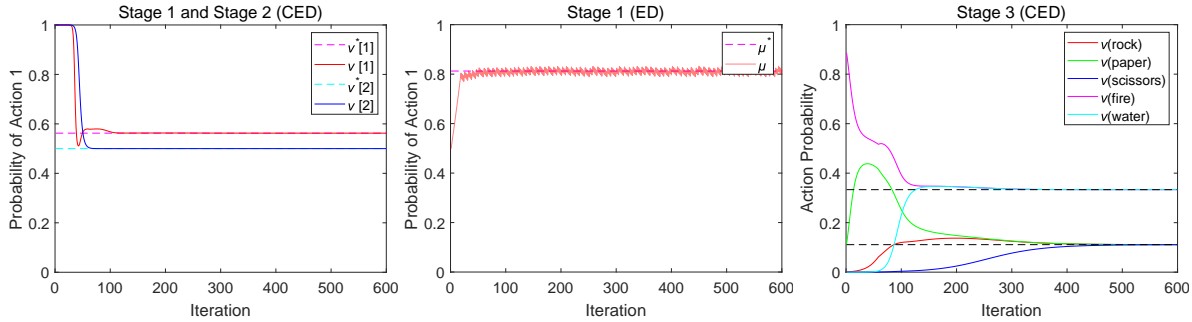

*Figure 2.* CED and ED learning curves in the tree-form game

consistent with Theorem 5.2 and Theorem 5.6, which claim that under uniform coverage, CED can converge to an unexploitable $\nu$ (an NE policy in this case). We may find that the learned $\mu$ for $\mathcal{M}_2$ also corresponds to the equilibrium. However, this is because $\nu_\beta$ happens to be $\nu^*$ in $\mathcal{M}_2$. Otherwise, the divergence regularization applied to the computation of $\nu$ will force the stationary point of $\mu$ to deviate from $\mu^*$ because $\nu^*$ is not an exact best response to the stationary $\mu$. This phenomenon is shown in the learning curve on $\mathcal{M}_1$, with the ultimate $\mu \neq \mu^*$ as a result of $\nu_\beta \neq \nu^*$.

We also test CED in a 5-action "Rock-Paper-Scissors-Fire-Water" game denoted by $\mathcal{M}_3$. Besides the common rules of the RPS game, fire beats everything except water, and water is beaten by everything except it beats fire. $\left(\frac{1}{9}, \frac{1}{9}, \frac{1}{9}, \frac{1}{3}, \frac{1}{3}\right)$ is an unexploitable policy for both players, and the unique Nash equilibrium of $\mathcal{M}_3$ is constructed when both players use this policy. As is shown in Figure 1 (right), CED ($\alpha = 0.01, \epsilon = 0.1$) manages to learn the mixed-strategy equilibrium policy, where the black dashed lines correspond to the action probabilities of $\frac{1}{3}$ and $\frac{1}{9}$, respectively.

## 6.2. Tree-Form Game

Now we further consider dynamic games, where the Nash equilibrium at a decision point is affected by the results of subsequent game stages. We examine the learning behaviors in a tree-form game $\mathcal{T}$ consisting of three decision points whose payoff matrices are $\mathcal{M}_1$, $\mathcal{M}_2$, and $\mathcal{M}_3$, respectively. $\mathcal{T}$ starts with Stage 1 ($\mathcal{M}_1$) and enters Stage 2 ($\mathcal{M}_2$) or Stage 3 ($\mathcal{M}_3$) conditioned on the joint actions of two players at Stage 1 (see Appendix B.1). By backward induction, we can compute that the NE at Stage 1 is $\left(\mu^*(1) = \frac{13}{16}, \nu^*(1) = \frac{9}{16}\right)$, which deviates from the original equilibrium point $\left(\frac{6}{7}, \frac{4}{7}\right)$ in $\mathcal{M}_1$.

As is shown in Figure 2 (left & right), CED ($\alpha = 0.005, \epsilon = 0.1$) finds the NE policy for the min-player in the tree-form game. As there is a mismatch between the convergence speed at Stage 2 and Stage 3, $\nu$ at Stage 1 experiences an oscillation and eventually converges to the solution. This phenomenon is consistent with the intuition that the learning process at the initial stage depends on subsequent stages in dynamic games. Besides, we test the behavior of the model-based ED algorithm in this scenario. As is shown in Figure 2 (mid), while ED can approximate the NE policy for the max-player, it suffers from continual oscillations as a side effect of following a generalized gradient.

## 6.3. Soccer Game

While the theoretical analysis and the toy problem experiments above have suggested the capability of CED to find mixed-strategy Nash equilibrium, here we further verify the conclusion in an infinite-horizon Markov game, the soccer game (see Appendix B.2). To measure the performance of CED, we compute the NashConv of the learned $(\mu, \nu)$

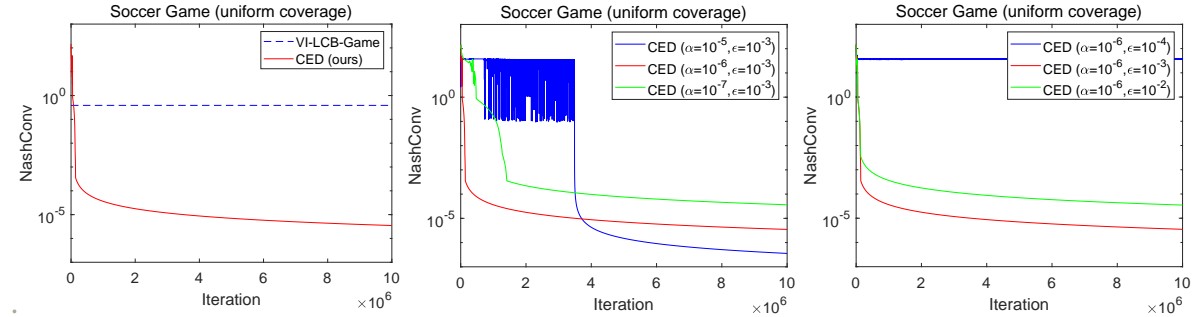

*Figure 3.* Learning curve comparisons under uniform coverage in the soccer game

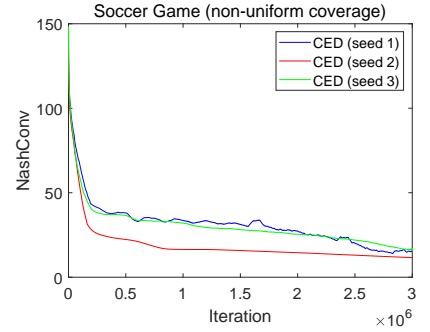

| Performance comparison between learned policy and behavior policy under different coverages | | |
|---|---|---|
| Player 1 | Player 2 | Win Rate |
| Learned policy from uniform coverage | Behavior policy from uniform coverage | 92.4% |
| | Behavior policy from non-uniform coverage | 91.9% |
| Learned policy from non-uniform coverage | Behavior policy from uniform coverage | 91.8% |
| | Behavior policy from non-uniform coverage | 91.8% |
| | Learned policy from uniform coverage | 45.8% |

*Figure 4.* Performance improvements over behavior policy by CED in the soccer game

and compare it with the result of a pessimistic model-based algorithm, VI-LCB-Game (Yan et al., 2024), which provably finds approximate Nash equilibrium offline for infinite-horizon MGs but requires infinitely many samples in theory. In Figure 3 (left), the dashed line corresponds to the joint policy derived from VI-LCB-Game, given the minimum amount of samples for uniform coverage. Under the same offline dataset, CED steadily reduces the exploitability of the learned policy in this symmetric MG and eventually obtains a policy with a significantly lower NashConv.

Theorem 5.2 proves the convergence of CED under a sufficiently small $\alpha$ and $\frac{1}{\epsilon}$. However, we are also curious about whether it is possible to use different $\alpha$ and $\epsilon$ in practice. As is shown in Figure 3 (mid), an overly large $\alpha$ makes it significantly harder for CED to converge, while an overly small $\alpha$ slows down the speed of learning. Figure 3 (right) shows that it does not affect convergence to use a small (but not too small) regularization parameter $\epsilon$. These results can be viewed as a supplement to our theoretical analysis.

As CED is model-free and does not rely on the full game information, it is, in principle, applicable to an arbitrary set of offline data, regardless of the coverage. Here we further examine if it can gradually improve the behavior policy when the coverage is non-uniform, like those single-agent offline RL algorithms. To be specific, we randomly banned one action out of five for each player at each state and removed

all the related transitions from the dataset $\mathcal{D}$. This makes it impossible to learn an exact Nash equilibrium in theory, as a preferred action from the NE can be completely removed. As is shown in Figure 4 (left), CED still gradually improves the behavior policy under such random data coverages.

Besides NashConv, we estimate the win rate to intuitively show the improvement over behavior policy by CED. As is shown in Figure 4 (right), whether under uniform or non-uniform coverage, the policy learned by CED significantly improves the practical performance, with win rates over 90% against the behavior policies. When playing against the approximate Nash equilibrium policy learned by CED from the dataset with uniform coverage, the learned policy from non-uniform coverage still achieves a close win rate (45.8%). As no policy can possibly win an NE policy with probability over 50% in a symmetric two-player zero-sum game, this result reflects that CED can learn a competitive policy even from the datasets without uniform coverage.

### 6.4. Robotic Combat Game

Now we show that CED can also be combined with neural networks to approximately solve large-scale adversarial games in an offline manner. We implement CED in a graph-based two-team robotic combat game (see Appendix B.3). We construct an offline dataset that contains 2000 game trajectories, where the actual behavior policy for both teams is

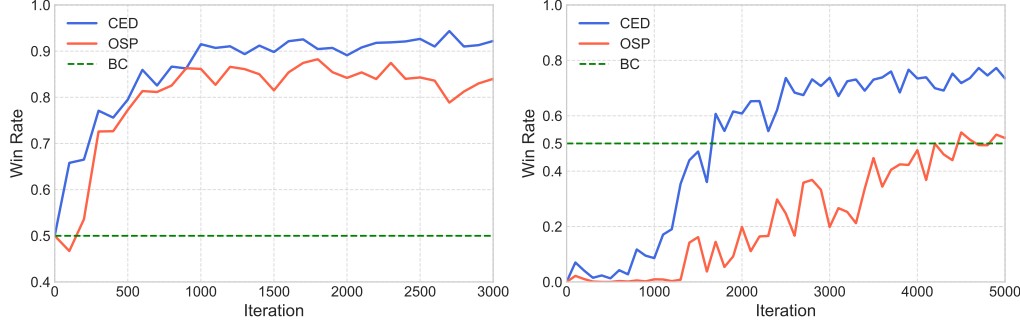

*Figure 5.* Learning curves under BC initialization (left) and random initialization (right) in the robotic combat game

| Iteration 3000 | CED (BC_init) | OSP (BC_init) | CED (RD_init) | OSP (RD_init) |
|---|---|---|---|---|
| CED (BC_init) | 50% | 52% | 66% | 67% |
| OSP (BC_init) | 48% | 50% | 55% | 65% |
| CED (RD_init) | 34% | 45% | 50% | 65% |
| OSP (RD_init) | 33% | 35% | 35% | 50% |

| Iteration 5000 | CED (BC_init) | OSP (BC_init) | CED (RD_init) | OSP (RD_init) |
|---|---|---|---|---|
| CED (BC_init) | 50% | 51% | 61% | 58% |
| OSP (BC_init) | 49% | 50% | 55% | 57% |
| CED (RD_init) | 39% | 45% | 50% | 57% |
| OSP (RD_init) | 42% | 43% | 43% | 50% |

*Figure 6.* Win rate comparisons of CED / OSP policy under BC / random initialization in the robotic combat game

a cooperative multi-agent policy previously trained against a rule-based opponent through online RL. We first use supervised behavior cloning (BC) to approximate the behavior policy from the dataset. This corresponds to the first step of CED (Algorithm 2). As we verify that the learned behavior policy has a win rate around 50% against the actual behavior policy, we simply use the win rate against the learned behavior policy from BC as the performance measure.

We compare the last-iterate performance of CED with offline self-play (OSP in Chen et al. (2024)) under the same network architecture that represents multi-agent policies on graphs for both teams. We also consider two initializations under behavior policy and random policy. As is shown in Figure 5, under either initialization, both CED and OSP eventually outperform BC-approximated behavior policy, and CED has a comparatively better learning performance than OSP. Figure 6 shows the tested win rates of the four learned policies against each other. While the initializations under behavior policy have a clear advantage over random initializations, randomly initialized CED still has a close win rate against BC-initialized OSP (45%) under the same number of iterations (3000 or 5000). That is to say, CED can consistently outperform BC and OSP methods.

# 7. Conclusion

In this paper, by proposing CED and analyzing its convergence properties, we demonstrate for the first time that, unlike in MDPs, an optimal policy can be learned under policy constraints in adversarial MGs. This conclusion is drawn from our theoretical and empirical results. With Theorem 5.2 and Theorem 5.6, we prove that under uniform coverage, CED converges to an unexploitable min-player policy without relying on the generalized gradient. In the experiments, our theory is verified by the practical results of CED in multiple game scenarios. We also show that, similar to single-agent offline RL algorithms, CED can improve the behavior policy from datasets without uniform coverage, even under function approximations for large-scale games.

We hope this work will inspire more research on solving offline games. Actually, since CED is constructed based on the game-theoretic approach of Exploitability Descent, which is also capable of solving imperfect-information games (IIGs), it is possible to use CED as an offline IIG solver by replacing the state and value with the information state and counterfactual value. However, how to estimate counterfactual value under the current policy using offline game data remains an open problem. Further theoretical analysis is still required.

CED has the theoretical limitation that it is only guaranteed to find mixed-strategy Nash equilibria in two-player zero-sum games. However, it may not be the unique way of equilibrium learning under policy constraints, as a wide range of algorithms that exhibit last-iterate convergence (e.g., DRDA (Lu et al., 2025)) are currently available in the field of game theory. Combining them with existing offline RL techniques may lead to more offline RL algorithms with possibly better guarantees to find Nash equilibrium.

## Acknowledgements

This work was supported in part by the National Natural Science Foundation of China under Grant 62293541 and Grant 62136008, in part by the Beijing Natural Science Foundation under Grant 4232056, and in part by the Beijing Nova Program under Grant 20240484514. We would like to thank the reviewers for providing valuable comments and Ruochuan Shi for helping to evaluate CED under function approximations in the robotic combat game.

## Impact Statement

This paper presents work whose goal is to advance the field of machine learning. There are many potential societal consequences of our work, and we feel that none of them must be specifically highlighted here.

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

# A. Omitted Proofs

### A.1. Proof of Lemma 4.1

*Proof.* First, we prove:

$$\pi = \arg\max_{\pi \in \Delta(\mathcal{A})} \left\{ \sum_{a \in \mathcal{A}} \pi(a) \left( r(a) - \log \pi(a) \right) \right\} \Rightarrow \pi(a) \propto e^{r(a)}$$

Write the corresponding optimization problem:

$$\begin{cases} \text{maximize } \displaystyle\sum_{a \in \mathcal{A}} \pi(a) \left( r(a) - \log \pi(a) \right) \\ \text{s.t. } \displaystyle\sum_{a \in \mathcal{A}} \pi(a) = 1 \\ \qquad \pi(a) \geq 0, \ \forall a \in \mathcal{A} \end{cases}$$

Using the Lagrange multiplier, we have:

$$L = \sum_{a \in \mathcal{A}} \pi(a) \left( r(a) - \log \pi(a) \right) - \lambda \left( \sum_{a \in \mathcal{A}} \pi(a) - 1 \right)$$

$$\frac{\partial L}{\partial \pi(a)} = 0 \Rightarrow r(a) - \left( \log \pi(a) + \frac{\pi(a)}{\pi(a)} \right) - \lambda = 0$$

$$\Rightarrow \pi(a) = e^{r(a) - \lambda - 1} \Rightarrow \pi(a) \propto e^{r(a)}$$

By definition of $\nu_k$, we have:

$$\nu_k(s) = \arg\max_{\nu(s) \in \Delta(\mathcal{B})} \left\{ \sum_{b \in \mathcal{B}} \nu(s, b) \left( - \sum_{a \in \mathcal{A}} \mu_k(s, a) Q^{\mu_\beta, \nu_\beta}(s, a, b) \right) - \epsilon D_{\mathrm{KL}} \left( \nu(s), \nu_\beta(s) \right) \right\}$$

$$= \arg\max_{\nu(s) \in \Delta(\mathcal{B})} \left\{ \sum_{b \in \mathcal{B}} \nu(s, b) \left( -\frac{1}{\epsilon} \sum_{a \in \mathcal{A}} \mu_k(s, a) Q^{\mu_\beta, \nu_\beta}(s, a, b) - \log \frac{\nu(s, b)}{\nu_\beta(s, b)} \right) \right\}$$

$$= \arg\max_{\nu(s) \in \Delta(\mathcal{B})} \left\{ \sum_{b \in \mathcal{B}} \nu(s, b) \left( \log \nu_\beta(s, b) - \frac{1}{\epsilon} \sum_{a \in \mathcal{A}} \mu_k(s, a) Q^{\mu_\beta, \nu_\beta}(s, a, b) - \log \nu(s, b) \right) \right\}$$

Therefore:

$$\nu_k(s, b) \propto \exp \left( \log \nu_\beta(s, b) - \frac{1}{\epsilon} \sum_{a \in \mathcal{A}} \mu_k(s, a) Q^{\mu_\beta, \nu_\beta}(s, a, b) \right)$$

which implies:

$$\nu_k(s, b) = \frac{\nu_\beta(s, b) \exp \left( -\frac{1}{\epsilon} \sum_{a \in \mathcal{A}} \mu_k(s, a) Q^{\mu_\beta, \nu_\beta}(s, a, b) \right)}{\sum_{b' \in \mathcal{B}} \nu_\beta(s, b') \exp \left( -\frac{1}{\epsilon} \sum_{a \in \mathcal{A}} \mu_k(s, a) Q^{\mu_\beta, \nu_\beta}(s, a, b') \right)}$$

$\square$

## A.2. Proof of Lemma 5.1

*Proof.* By definition:

$$
\begin{aligned}
\frac{\partial V^{\mu,\nu}(s)}{\partial \mu(\hat{s},a)} &= \frac{\partial}{\partial \mu(\hat{s},a)} \sum_{a\in\mathcal{A}} \mu(s,a) \sum_{b\in\mathcal{B}} \nu(s,b) Q^{\mu,\nu}(s,a,b) \\
&= \sum_{a\in\mathcal{A}} \left( \frac{\partial \mu(s,a)}{\partial \mu(\hat{s},a)} \sum_{b\in\mathcal{B}} \nu(s,b) Q^{\mu,\nu}(s,a,b) + \mu(s,a) \sum_{b\in\mathcal{B}} \nu(s,b) \frac{\partial Q^{\mu,\nu}(s,a,b)}{\partial \mu(\hat{s},a)} \right) \\
&= \mathbb{I}[s=\hat{s}] \sum_{b\in\mathcal{B}} \nu(s,b) Q^{\mu,\nu}(s,a,b) + \mu(s,a) \sum_{b\in\mathcal{B}} \nu(s,b) \frac{\partial}{\partial \mu(\hat{s},a)} \left( r(s,a,b) + \gamma V^{\mu,\nu}(s') \right) \\
&= \mathbb{I}[s=\hat{s}] \sum_{b\in\mathcal{B}} \nu(s,b) Q^{\mu,\nu}(s,a,b) + \sum_{a\in\mathcal{A}} \mu(s,a) \sum_{b\in\mathcal{B}} \nu(s,b) \gamma \frac{\partial V^{\mu,\nu}(s')}{\partial \mu(\hat{s},a)} \\
&= \cdots\cdots \\
&= \sum_{k=0}^{\infty} \gamma^k \Pr(s \to \hat{s}|k; \mu, \nu) \sum_{b\in\mathcal{B}} \nu(\hat{s},b) Q^{\mu,\nu}(\hat{s},a,b)
\end{aligned}
$$

where $\mathbb{I}[\cdot]$ is the indicator function and $\Pr(s \to \hat{s}|k; \mu, \nu)$ is the probability of reaching $\hat{s}$ from $s$ using $k$ steps under joint policy $(\mu, \nu)$.

Then, it is direct to show:

$$
\begin{aligned}
\frac{\partial u(\mu,\nu)}{\partial \mu(s,a)} &= \frac{\partial}{\partial \mu(s,a)} \mathbb{E}_{s_0 \sim \rho_0} \left[ V^{\mu,\nu}(s_0) \right] \\
&= \sum_{s_0 \in S} \rho_0(s_0) \sum_{k=0}^{\infty} \gamma^k \Pr(s_0 \to s|k; \mu, \nu) \sum_{b\in\mathcal{B}} \nu(s,b) Q^{\mu,\nu}(s,a,b) \\
&= \sum_{k=0}^{\infty} \gamma^k \Pr(s|k; \mu, \nu) \sum_{b\in\mathcal{B}} \nu(s,b) Q^{\mu,\nu}(s,a,b) \\
&= \rho^{\mu,\nu}(s) \sum_{b\in\mathcal{B}} \nu(s,b) Q^{\mu,\nu}(s,a,b)
\end{aligned}
$$

$\square$

### A.3. Details in the Proof of Theorem 5.2

Here, we will show that $\frac{\partial \nu_k(s,b)}{\partial \mu_k(s,a)} \to 0$ when $\frac{1}{\epsilon} \to 0$.

By Lemma 4.1:

$$\nu_k(s,b) = \frac{\nu_\beta(s,b)\exp\left(-\frac{1}{\epsilon}\sum_{a\in\mathcal{A}}\mu_k(s,a)Q^{\mu_\beta,\nu_\beta}(s,a,b)\right)}{\sum_{b'\in\mathcal{B}}\nu_\beta(s,b')\exp\left(-\frac{1}{\epsilon}\sum_{a\in\mathcal{A}}\mu_k(s,a)Q^{\mu_\beta,\nu_\beta}(s,a,b')\right)}$$

Besides:

$$\frac{\partial\exp\left(-\frac{1}{\epsilon}\sum_{a\in\mathcal{A}}\mu_k(s,a)Q^{\mu_\beta,\nu_\beta}(s,a,b)\right)}{\partial\mu_k(s,a)} =$$

$$-\frac{1}{\epsilon}Q^{\mu_\beta,\nu_\beta}(s,a,b)\exp\left(-\frac{1}{\epsilon}\sum_{a\in\mathcal{A}}\mu_k(s,a)Q^{\mu_\beta,\nu_\beta}(s,a,b)\right)$$

Therefore:

$$\frac{\partial\nu_k(s,b)}{\partial\mu_k(s,a)} = \frac{1}{\epsilon}\nu_\beta(s,b)\exp\left(-\frac{1}{\epsilon}\sum_{a\in\mathcal{A}}\mu_k(s,a)Q^{\mu_\beta,\nu_\beta}(s,a,b)\right)\cdot$$

$$\frac{\sum_{b'\in\mathcal{B}}\nu_\beta(s,b')\exp\left(-\frac{1}{\epsilon}\sum_{a\in\mathcal{A}}\mu_k(s,a)Q^{\mu_\beta,\nu_\beta}(s,a,b')\right)\left(Q^{\mu_\beta,\nu_\beta}(s,a,b')-Q^{\mu_\beta,\nu_\beta}(s,a,b)\right)}{\left(\sum_{b'\in\mathcal{B}}\nu_\beta(s,b')\exp\left(-\frac{1}{\epsilon}\sum_{a\in\mathcal{A}}\mu_k(s,a)Q^{\mu_\beta,\nu_\beta}(s,a,b')\right)\right)^2}$$

Now, it is clear:

$$\lim_{\frac{1}{\epsilon}\to 0}\frac{\partial\nu_k(s,b)}{\partial\mu_k(s,a)} = 0\cdot\frac{\sum_{b'\in\mathcal{B}}\nu_\beta(s,b')\left(Q^{\mu_\beta,\nu_\beta}(s,a,b')-Q^{\mu_\beta,\nu_\beta}(s,a,b)\right)}{\left(\sum_{b'\in\mathcal{B}}\nu_\beta(s,b')\right)^2} = 0$$

### A.4. Proof of Lemma 5.4

*Proof.* Without loss of generality, we prove the first half that $\mu^*$ is unexploitable with respect to $\nu^*$. We show that $\sum_{a \in \mathcal{A}} \mu^*(s,a) Q^{\mu^*,\nu^*}(s,a,b_1) > \sum_{a \in \mathcal{A}} \mu^*(s,a) Q^{\mu^*,\nu^*}(s,a,b_2)$ leads to a contradiction when $(\mu^*, \nu^*)$ is a Nash equilibrium with full support. By definition, the value at state $s$ is:

$$V^{\mu^*,\nu^*}(s) = \sum_{b \in \mathcal{B}} \nu^*(s,b) \sum_{a \in \mathcal{A}} \mu^*(s,a) Q^{\mu^*,\nu^*}(s,a,b)$$

When $\nu^*(s)$ has nonzero probability at each $b \in \mathcal{B}$, decreasing $\nu^*(s, b_1)$ and increasing $\nu^*(s, b_2)$ should decrease the value for the min-player. Therefore, $\nu^*$ is not a best response against $\mu^*$, which contradicts the NE assumption. $\qquad\square$

### A.5. Proof of Lemma 5.5

*Proof.* By definition:

$$\sum_{a \in \mathcal{A}} \left( \mu(s,a) - (\mu_k(s,a) + z_a^s) \right)^2$$

$$= \sum_{a \in \mathcal{A}} \left( \mu(s,a) - \left( \mu_k(s,a) + p_a^s + \frac{y}{|\mathcal{A}|} \right) \right)^2$$

$$= \sum_{a \in \mathcal{A}} \left( (\mu(s,a) - (\mu_k(s,a) + p_a^s)) - \frac{y}{|\mathcal{A}|} \right)^2$$

$$= \sum_{a \in \mathcal{A}} (\mu(s,a) - (\mu_k(s,a) + p_a^s))^2 + \sum_{a \in \mathcal{A}} \left( \frac{y}{|\mathcal{A}|} \right)^2 - \frac{2y}{|\mathcal{A}|} \sum_{a \in \mathcal{A}} (\mu(s,a) - (\mu_k(s,a) + p_a^s))$$

$$= \sum_{a \in \mathcal{A}} (\mu(s,a) - (\mu_k(s,a) + p_a^s))^2 + \frac{y^2}{|\mathcal{A}|} - \frac{2y}{|\mathcal{A}|} \left( \sum_{a \in \mathcal{A}} \mu(s,a) - \sum_{a \in \mathcal{A}} \mu_k(s,a) + \sum_{a \in \mathcal{A}} z_a^s - \sum_{a \in \mathcal{A}} \frac{\sum_{a \in \mathcal{A}} z_a^s}{|\mathcal{A}|} \right)$$

$$= \sum_{a \in \mathcal{A}} (\mu(s,a) - (\mu_k(s,a) + p_a^s))^2 + \frac{y^2}{|\mathcal{A}|} - \frac{2y}{|\mathcal{A}|} (1 - 1)$$

$$= \sum_{a \in \mathcal{A}} (\mu(s,a) - (\mu_k(s,a) + p_a^s))^2 + \frac{y^2}{|\mathcal{A}|}$$

Therefore:

$$\mu_{k+1}(s) = \operatorname*{arg\,min}_{\mu(s) \in \Delta(\mathcal{A})} \sum_{a \in \mathcal{A}} (\mu(s,a) - (\mu_k(s,a) + z_a^s))^2 = \operatorname*{arg\,min}_{\mu(s) \in \Delta(\mathcal{A})} \sum_{a \in \mathcal{A}} (\mu(s,a) - (\mu_k(s,a) + p_a^s))^2$$

$\qquad\square$

## A.6. Policy Penalty Bound

We use the following lemma to rigorously demonstrate that the indirect policy penalty in CED can bound the distance between the learned policy $\nu_k$ and the behavior policy $\nu_\beta$.

**Lemma A.1** (Policy Penalty Bound). *Let $Q_{\max}$ and $Q_{\min}$ be the maximum and minimum values of $Q^{\mu_\beta, \nu_\beta}$ and let $C > 0$ be any threshold. When $\epsilon \geq \frac{Q_{\max} - Q_{\min}}{\log(1 + C)}$, it holds that $\|\nu_k(s) - \nu_\beta(s)\|_1 \leq C$ for all $s \in S$ in the CED algorithm.*

*Proof.* By Lemma 4.1, we have:

$$\nu_k(s, b) = \frac{\nu_\beta(s, b) \exp \left( -\frac{1}{\epsilon} \sum_{a \in \mathcal{A}} \mu_k(s, a) Q^{\mu_\beta, \nu_\beta}(s, a, b) \right)}{\sum_{b' \in \mathcal{B}} \nu_\beta(s, b') \exp \left( -\frac{1}{\epsilon} \sum_{a \in \mathcal{A}} \mu_k(s, a) Q^{\mu_\beta, \nu_\beta}(s, a, b') \right)}$$

Let $t = \frac{\nu_\beta(s, b)}{\nu_k(s, b)} = \sum_{b' \in \mathcal{B}} \nu_\beta(s, b') \exp \left( \frac{1}{\epsilon} \sum_{a \in \mathcal{A}} \mu_k(s, a) \left( Q^{\mu_\beta, \nu_\beta}(s, a, b) - Q^{\mu_\beta, \nu_\beta}(s, a, b') \right) \right)$.

By definition of $Q_{\max}$ and $Q_{\min}$, we have:

$$Q_{\min} - Q_{\max} \leq Q^{\mu_\beta, \nu_\beta}(s, a, b) - Q^{\mu_\beta, \nu_\beta}(s, a, b') \leq Q_{\max} - Q_{\min}$$

Since $\sum_{a \in \mathcal{A}} \mu_k(s, a) = 1$, we have:

$$\frac{Q_{\min} - Q_{\max}}{\epsilon} \leq \frac{1}{\epsilon} \sum_{a \in \mathcal{A}} \mu_k(s, a) \left( Q^{\mu_\beta, \nu_\beta}(s, a, b) - Q^{\mu_\beta, \nu_\beta}(s, a, b') \right) \leq \frac{Q_{\max} - Q_{\min}}{\epsilon}$$

Since $\sum_{b' \in \mathcal{B}} \nu_\beta(s, b') = 1$, we further have:

$$\exp \left( \frac{Q_{\min} - Q_{\max}}{\epsilon} \right) \leq t \leq \exp \left( \frac{Q_{\max} - Q_{\min}}{\epsilon} \right)$$

Since $\epsilon \geq \frac{Q_{\max} - Q_{\min}}{\log(1 + C)}$, it holds that $\exp \left( \frac{Q_{\max} - Q_{\min}}{\epsilon} \right) \leq 1 + C$. Therefore, $t \leq 1 + C$.

When $C \geq 1$, it is clear that $\exp \left( \frac{Q_{\min} - Q_{\max}}{\epsilon} \right) \geq 1 - C$. When $0 < C < 1$, we have:

$$\epsilon \geq \frac{Q_{\max} - Q_{\min}}{\log(1 + C)} \geq \frac{Q_{\max} - Q_{\min}}{-\log(1 - C)} = \frac{Q_{\min} - Q_{\max}}{\log(1 - C)}$$

It is also clear that $\exp \left( \frac{Q_{\min} - Q_{\max}}{\epsilon} \right) \geq 1 - C$. Therefore, $t \geq 1 - C$.

Since $|\nu_k(s, b) - \nu_\beta(s, b)| = |\nu_k(s, b)(1 - t)| \leq \nu_k(s, b) |1 - t|$, we have:

$$\|\nu_k(s) - \nu_\beta(s)\|_1 \leq \sum_{b \in \mathcal{B}} \nu_k(s, b) |1 - t| = |1 - t| \leq C$$

$\square$

# B. Test Environments

### B.1. Tree-Form Game

We use a tree-form game $\mathcal{T}$ as a test environment for both CED and ED algorithms. Figure 7 is an illustration of $\mathcal{T}$, which consists of three decision points with payoff matrices $\mathcal{M}_1$, $\mathcal{M}_2$, and $\mathcal{M}_3$, respectively. $\mathcal{T}$ starts with Stage 1 ($\mathcal{M}_1$) and enters Stage 2 ($\mathcal{M}_2$) or Stage 3 ($\mathcal{M}_3$) conditioned on previous actions. If both use the same action 0 or 1, $\mathcal{T}$ enters Stage 2. Otherwise, $\mathcal{T}$ enters Stage 3.

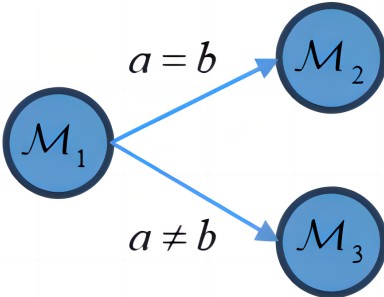

Figure 7. Illustration of tree-form game

### B.2. Soccer Game

We use a two-player zero-sum soccer game as a small-scale environment for infinite-horizon MGs. Figure 8 is an illustration of the game. The two players are marked with A and B. The player who keeps the ball is marked with a circle. Each player can choose an action from "up", "down", "left", "right", and "stay" at each time step. If the two players collide after the simultaneous move, then the ball possession exchanges. When the ball carrier moves into the opponent's goal, the game terminates. The winning player receives a reward of $+100$, and the opponent receives a reward of $-100$. The initial state distribution $\rho_0$ is set to be uniform, and the discount factor $\gamma$ is set to be 0.95.

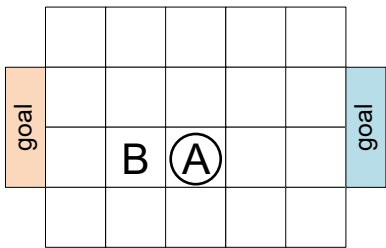

Figure 8. Illustration of soccer game

### B.3. Robotic Combat Game

We use a two-team zero-sum robotic combat game as an environment for large-scale MGs. Figure 9 is an illustration of the game, where each team consists of three homogeneous robots. The game map is abstracted as a 100-node graph, where each robot can move to an unoccupied adjacent node or attack an enemy at each time step. The HP reduction is influenced by the terrain and actual distance between the attacker and the target. The robots under different terrains cannot damage each other (e.g., green cannot damage purple but can damage orange since nodes 55 and 72 are at the same level), and the damage to node 55 from node 61 is much higher than from node 72. A team of robots wins if all opponents' HP is reduced to zero.

# C. Further Explanations of CED

Recall that the NE strategy $\mu^*$ for the max-player always satisfies $\mu^* = \arg\max_\mu \{\min_\nu u(\mu, \nu)\}$. The idea of ED is to update $\mu$ along the gradient of $\min_\nu u(\mu, \nu)$. However, this gradient may not exist since $\text{br}(\mu) := \arg\min_\nu u(\mu, \nu)$ may have multiple solutions. Therefore, by fixing an arbitrary $\nu' \in \text{br}(\mu)$, a generalized gradient $\frac{\partial u(\mu, \nu')}{\partial \mu} \in \partial \min_\nu u(\mu, \nu)$ is

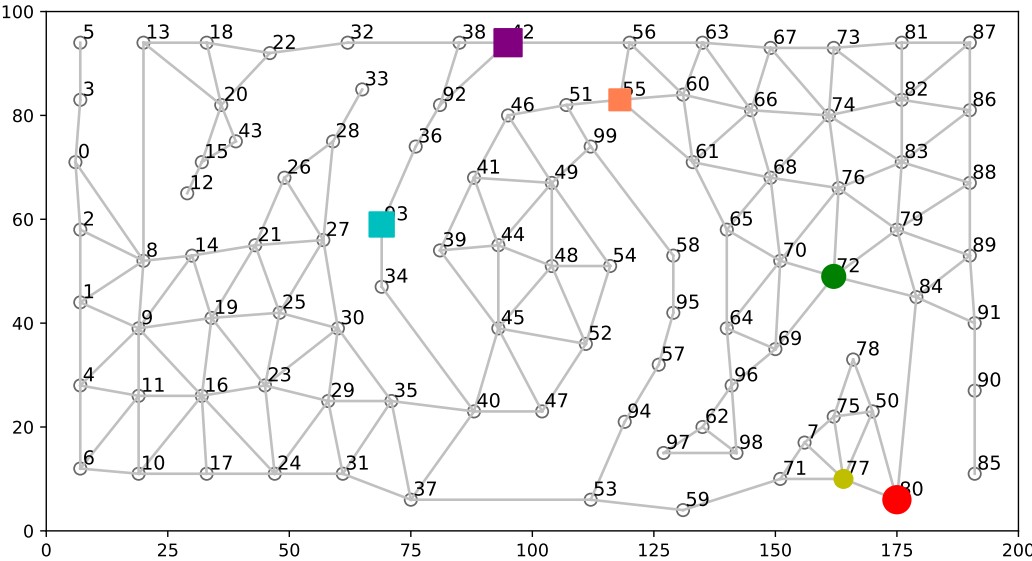

*Figure 9.* Illustration of robotic combat game (Team Circle vs Team Square)

used instead. Lockhart et al. (2019) prove that the best iterate of the max-player policy $\mu$ can converge to a Nash equilibrium.

For CED (Algorithm 2), since the computation of $\nu$ is under divergence regularization (indirect policy constraint), it is uniquely determined by $\mu$ but is no longer an exact best response to $\mu$. The benefit is that we can directly prove *last-iterate convergence* (see Lu et al. (2024)) rather than best-iterate convergence of $\mu$. The problem is that the update of $\mu$ does not follow a gradient induced by its best response and cannot converge to the NE strategy. However, as long as the limit point is interior in the constrained policy set, we can use the projected update formula in Lemma 5.5 to prove that $\mu$ has the same value for all actions at any given state $s \in S$. Therefore, the min-player policy $\nu$ satisfies the property of mixed-strategy NE, i.e., being unexploitable with respect to its opponent policy. The NE policies in our matrix/tree-form game experiments are the explicit examples. Note that ED itself does not have this property because the learned policy is unstable around a local optimum of the minimax problem. From the perspective of offline RL, the policy constraints in CED also mitigate the problem of encountering out-of-distribution states and actions, guaranteeing a bounded distributional shift.

Besides, directly extending ED to deep RL algorithms faces the problem that we need to approximate a best response (BR) of the current $\mu$ in every single gradient update. However, this requirement is relaxed in CED (Algorithm 2). Actually, the "approximate best response" that we require in the second inner loop is only at the level of each single state, under a state-action value function $Q^{\mu_\beta, \nu_\beta}$ preprocessed outside the main loop. This is in sharp contrast to computing an exact BR against the current $\mu$ and does not need a separate BR oracle at all. In our tabular experiments, we simply traverse all states and compute the current $\nu$ by Lemma 4.1. When we employ a function approximator for $\nu$, a direct approach is to update its parameters along the gradient of the current target in the first inner loop. Since this target only changes with $\mu(s)$ at each state $s \in S$, it is reasonable for $\nu$ to take a comparative amount of gradient steps as $\mu$ in each iteration of the main loop.

## D. Parameter Selection Details

For the learning rate $\alpha$, Theorem 5.2 provides a guideline that it should be sufficiently small. However, an overly small $\alpha$ will slow down the speed of convergence, as is shown in Figure 3 (mid). Therefore, there is a trade-off with respect to the selection of $\alpha$. For the policy penalty parameter $\epsilon$, Theorem 5.2 also provides a guideline that it should not be overly small, as is verified in Figure 3 (right). However, it is risky to set an overly large $\epsilon$ because the interior point condition in Theorem 5.6 is implicitly affected by the policy constraint on the min-player policy $\nu$. Therefore, there is also a trade-off with respect to the selection of $\epsilon$. From our experience, as long as $\epsilon$ is sufficiently large for the practical convergence of CED, a relatively small $\epsilon$ generally guarantees a relatively low NashConv of the learned policy $\nu$ under the same number of iterations.

