# OpenReview forum: "Constrained Exploitability Descent: An Offline Reinforcement Learning Method for Finding Mixed-Strategy Nash Equilibrium"
_ICML.cc/2025/Conference — ICML 2025 poster_

### Official Review · Reviewer_shnB · 2025-03-12

**Overall Recommendation:** 3

**Summary:**

This paper proposes an offline RL method to solve mix-strategy Nash Equilibrium via a game-theoretic method, exploitability descent.

**Claims And Evidence:**

Is best-iterate convergence better than average-iterate convergence? Would be good to get more detailed comments on this.

**Essential References Not Discussed:**

Some other offline RL for game solving literatures:

Li, Shuxin, et al. "Offline equilibrium finding." arXiv preprint arXiv:2207.05285 (2022).

Tang, Xiaohang, et al. "Adversarially Robust Decision Transformer." *arXiv preprint arXiv:2407.18414* (2024).

Chen, Jingxiao, et al. "Offline Fictitious Self-Play for Competitive Games." *arXiv preprint arXiv:2403.00841* (2024).

**Experimental Designs Or Analyses:**

Fig 2 is relatively confusing. I cannot conclude by looking at the dynamics of action probabilities without comparing to the solution of the game. Shouldn't the \mu^* and \nu^* plotted in the figure as in Fig 1?

**Methods And Evaluation Criteria:**

The main concern is the algorithmic design in Algo 2. From my understanding, we should compute exploitability first and then do descent of it. So \nu_0 shouldn't be \nu_\beta, but br(\mu_0). In practice it might not influence a lot, but I wonder if it is a simple mistake or the authors do it on purpose?

Evaluation demonstrates the convergence to mixed-strategy NE. But the results in Fig 2 to me did not show much gap from model-based ED.

Given ED is originally widely evaluated on pokers, it would be more convincing to test on pokers. Current environments are relatively toy.

Some baselines are lacking. To me, the proposed will be more convincing if it can beat behavior cloning. In those perfect-information games, it would be better to compare with simple in-sample minimax Q-learning (like multi-agent version of Implicit Q-Learning) introduced in this work:
Tang, Xiaohang, et al. "Adversarially Robust Decision Transformer." *arXiv preprint arXiv:2407.18414* (2024).

**Other Comments Or Suggestions:**

NA

**Other Strengths And Weaknesses:**

Strength:

This paper studies the open problem of offline equilibrium finding, leverage exploitability descent is an interesting direction.



Weakness:

The main issue is the insufficient environment and baselines. I'm happy to increase the score if the concerns are addressed.

**Questions For Authors:**

Is Markov Game a good formulation? For matrix game it is fine since it's state free. But tree-form game (extensive-form game) is history-based. In this case, will MG be a limited formulation?

**Relation To Broader Scientific Literature:**

This paper extend the following paper to offline RL setting:
Lockhart, Edward, et al. "Computing approximate equilibria in sequential adversarial games by exploitability descent." arXiv preprint arXiv:1903.05614 (2019).

**Theoretical Claims:**

In general make sense. Minor one: The notation of policy is problematic. In Algorithm 1 and 2, the authors have \mu sometimes a function of s, sometimes of s and a. Would be better to be rigorous.

---

> ### Author Rebuttal · Authors · 2025-04-01
>
> Thank you for reviewing this paper and providing valuable comments. We are updating the manuscript according to the comments from all reviewers. Here, we reply to your questions and concerns about the paper.
>
> **[Claims And Evidence]**
>
> Yes, average-iterate convergence means that we have to preserve an averaged policy from history policies. When the policy is represented by neural networks, such an averaging can hardly be accurate if we only preserve the parameters for the current policy. Even if we save the models of all history policies, it is costly to generate the average policy by querying each one of them. In comparison, best-iterate convergence usually has a similar behavior as last-iterate convergence, with a near-monotone policy improvement over training iterations. Under last-iterate convergence (which we prove for CED), it is reasonable to only preserve the current policy.
>
> **[Methods And Evaluation Criteria]**
>
> _**Comment**_: The main concern is the algorithmic design in Algo 2. From my understanding, we should compute exploitability first and then do descent of it. So \nu_0 shouldn't be \nu_\beta, but br(\mu_0). In practice it might not influence a lot, but I wonder if it is a simple mistake or the authors do it on purpose?
>
> Yes, we do it on purpose. Actually, a major technical difference between CED and ED is that CED optimizes the min-player strategy $\nu$ rather than the max-player strategy $\mu$. The purpose of CED is not to guarantee that $\mu$ approaches Nash equilibrium like ED, which is theoretically impossible when $\nu$ is regularized. Instead, we prove the unexploitability of the last-iterate $\nu$ under the convergence of $\mu$, with a theoretical analysis quite different from common best-iterate analysis on exploitability descent. Besides, CED does not compute an exact best response br$(\mu)$, but a state-level approximate BR in the last line of Algorithm 2. Since CED optimizes $\nu$, its computation is placed after $\mu$.
>
> _**Comment**_: Some baselines are lacking. To me, the proposed will be more convincing if it can beat behavior cloning.
>
> Actually, behavior cloning (BC) can serve as the first step of CED, i.e., to compute the behavior policy $(\mu_\beta,\nu_\beta)$ from the dataset. Since CED initializes $(\mu,\nu)$ to be $(\mu_\beta,\nu_\beta)$, the performance of the behavior policy corresponds to the starting points in Figures 3 and 4. In the tabular case, our existing experiments verify that CED consistently improves the performance of the behavior policy, which is in theory close to the result of BC.
>
> Currently, we have also implemented CED in a large-scale perfect-information game that simulates a two-team robotic combat scenario, where each team consists of three homogeneous robots. The game map is abstracted as a 100-node graph, where each robot can move to an unoccupied adjacent node or attack an enemy at each time step. A GIF illustration for a complete game is provided in the anonymous link https://sites.google.com/view/icml-2025-9335/.
>
> Among the mentioned references, Tang et al. (2024) require separate data relabeling and decision transformer training. Li et al. (2022) and Chen et al. (2024) employ PSRO and FSP, respectively, both of which require preserving all history policies. For a direct comparison, we alternatively use offline self-play (OSP in Chen et al.) as a baseline to test the last-iterate performance of CED under the same offline dataset (with 1000 trajectories) and network architecture (for representing the in-team joint policies). We consider two initializations with BC-approximated behavior policy or random policy. Figures A2 and A3 in the link show that CED can defeat BC policy under either initialization and has a comparatively better offline learning performance than OSP.
>
> **[Theoretical Claims]**
>
> Thanks for the comment. For the notation $\mu(s,a)$, it is defined as the probability of selecting action $a$ under policy $\mu(s)$ (in Line 125 of Page 3).
>
> **[Experimental Designs Or Analyses]**
>
> Lines 362-365 on Page 7 suggest that $\mu^*=\nu^*=(1/9,/1/9,1/9,1/3,1/3)$, and the dashed lines in Fig 2 correspond to $1/9$ and $1/3$, respectively. We will make it clearer in our revision.
>
> **[Questions For Authors]**
>
> As is stated in the conclusion part, we agree that it is a good direction to extend CED to imperfect-information games like poker. However, as shown in our response to the other two reviewers, CED has a few technical differences with ED and does not naturally follow the applicability of ED. In terms of offline learning, current theoretical works mainly focus on MGs and have few guarantees in imperfect-information games like history-based extensive-form games (EFGs). While we agree that MG is a simplified formulation, it is still challenging to answer fundamental problems like how to estimate the counterfactual value using offline game data in EFGs.
>
> Thanks again for your comments. We are looking forward to having further discussions with you.

---

> > ### Comment · Reviewer_shnB · 2025-04-02
> >
> > Thanks for the response. I have increased the score. If Figures 3 and 4 have BC, please mark it and add it to the legend for easier comparison.

---

> > > ### Author Response · Authors · 2025-04-02
> > >
> > > Thank you! We have marked BC in our updated figures for a direct comparison.

---

### Official Review · Reviewer_CuY7 · 2025-03-13

**Overall Recommendation:** 4

**Summary:**

The authors extend Exploitability Descent to the offline setting by applying a regularization constraint to minimize distance to the behavior policy. They provide theoretical guarantees for convergence under uniform concentration assumptions, and they provide experiments empirically validate their method, CED, on toy games.

**Claims And Evidence:**

The authors both prove and demonstrate CED's convergence properties under uniform concentration assumptions, and they also include an empirical result showing improvement over the behavior policy with non-uniform coverage.

**Essential References Not Discussed:**

The literature review is sufficient and accurately contextualizes this work in the broader field.

**Ethical Review Concerns:**

I have no ethical concerns for this work.

**Experimental Designs Or Analyses:**

The experimental design is sound. Leaving out an action at every state is a reasonable choice for the non-uniform coverage experiment.

**Methods And Evaluation Criteria:**

The evaluation criteria (demonstrating convergence to optimal actions and reduction of NashConv) with and without uniform dataset coverage are appropriate evaluation criteria.

However, the games tested are incredibly small. The authors argue in appendix C.2 that larger scale games are not evaluated on because calculating approximate NashConv with RL is an inaccurate measurement to use. I disagree with this argument under certain conditions. I believe this measurement accuracy tradeoff is acceptable if they were to scale up to slightly larger games in which RL algorithms can still reliably find approximately optimal solutions.

**Other Comments Or Suggestions:**

The y-axes for Figure 2 need to be labelled. More descriptive figure captions in general would improve readability greatly.

**Other Strengths And Weaknesses:**

Strengths:
- Extending online Markov game algorithms to the offline setting has immediate and clear utility for the community as a whole.
- The paper is well-written, and the claims made are reasonable and validated.

Weaknesses:
- My main complaint with this paper is that only toy games were tested on, and the authors did not consider an extension of CED to function approximation. I think doing so would have made the paper significantly stronger.
- I also am concerned with the scalability of CED (or future extensions of it) to larger games, and I ask the authors to address this in Questions.

**Questions For Authors:**

A critical limitation to scaling up ED with neural networks compared to methods like PSRO[1], MMD[2], and NFSP[3], is that an approximate best response operator would be required for every single gradient update. Wouldn't this easily cripple the usability of ED-based methods in larger games? Could the authors please comment on the severity of this limitation and discuss approaches the field has made towards addressing this? I think adding a paragraph on this would address a blind spot in the limitations.

[1] Lanctot, Marc, et al. "A unified game-theoretic approach to multiagent reinforcement learning."

[2] Sokota, Samuel, et al. "A Unified Approach to Reinforcement Learning, Quantal Response Equilibria, and Two-Player Zero-Sum Games."

[3] Heinrich, Johannes, and David Silver. "Deep reinforcement learning from self-play in imperfect-information games.

**Relation To Broader Scientific Literature:**

Most model-free methods for Markov games consider the online setting. Here, they extend Exploitability descent to the offline setting with Constrained Exploitability Descent.

**Theoretical Claims:**

I did not rigorously check the correctness of proofs.

---

> ### Author Rebuttal · Authors · 2025-04-01
>
> Thank you for reviewing this paper and providing valuable comments. We are updating the manuscript according to the comments from all reviewers. Here, we reply to your questions and concerns about the paper.
>
> **[Questions For Authors]**
>
> _**Question**_: A critical limitation to scaling up ED with neural networks compared to methods like PSRO[1], MMD[2], and NFSP[3], is that an approximate best response operator would be required for every single gradient update. Wouldn't this easily cripple the usability of ED-based methods in larger games? Could the authors please comment on the severity of this limitation and discuss approaches the field has made towards addressing this? I think adding a paragraph on this would address a blind spot in the limitations.
>
> We agree that directly extending ED to deep RL algorithms faces the problem that we need to approximate a best response (BR) of the current $\mu$ in every single gradient update. However, this requirement is relaxed in the proposed method CED (Algorithm 2). Actually, the "approximate best response" that we require in the second inner loop is only at the level of each single state, given a state-action value function $Q^{\mu_\beta,\nu_\beta}$ preprocessed outside the main loop. This is in sharp contrast to computing an exact BR against current $\mu$ and does not need a separate BR oracle at all. In our tabular experiments, we simply traverse all states and compute current $\nu$ by Lemma 4.1. When we employ a function approximator for $\nu$, a direct approach is to update its parameters along the gradient of the current target in Line 191. Since this target only changes with $\mu(s)$ at each state $s\in S$, it is reasonable for $\nu$ to take a comparative amount of gradient steps as $\mu$ in each iteration of the main loop. We can add a paragraph and discuss the benefit of this difference for potential deep RL extensions.
>
> **[Other Strengths And Weaknesses]**
>
> _**Comment**_: My main complaint with this paper is that only toy games were tested on, and the authors did not consider an extension of CED to function approximation. I think doing so would have made the paper significantly stronger.
>
> Thank you for this comment, and we agree that it is more convincing to evaluate CED under function approximation. Currently, we have implemented CED in a large-scale Markov game that simulates a two-team robotic combat scenario, where each team consists of three homogeneous robots. The game map is abstracted as a 100-node graph, where each robot can move to an unoccupied adjacent node or attack an enemy at each time step. The HP reduction is influenced by the terrain and actual distance between the attacker and the target. A GIF illustration for a complete game is provided in the anonymous link https://sites.google.com/view/icml-2025-9335/.
>
> We construct an offline dataset that contains $2000$ game trajectories, where the actual behavior policy for both teams is a cooperative MARL policy previously trained against a rule-based opponent. We first use supervised behavior cloning (BC) to approximate the behavior policy from the dataset. This corresponds to the first step of CED (Algorithm 2), and we verify that the learned behavior policy has a win rate around 50% against the actual behavior policy. Therefore, we simply use the win rate against the learned behavior policy (i.e., the result of behavior cloning) as the performance measure.
>
> We compare the last-iterate performance of CED with offline self-play (OSP in [4]) under the same network architecture. We consider initializations under either behavior policy or random policy. Figure A2 in the link shows that, under either initialization, both CED and OSP eventually outperform BC-approximated behavior policy, and CED has a comparatively better learning performance than OSP. Figure A3 shows the tested win rates of the four learned policies against each other. While the initializations under behavior policy have a clear advantage over random initializations, random-initialized CED still has a close win rate against BC-initialized OSP under the same number of iterations.
>
> **[Other Comments Or Suggestions]**
>
> _**Comment**_: The y-axes for Figure 2 need to be labelled. More descriptive figure captions in general would improve readability greatly.
>
> Thank you for this comment. In our revision, we have labeled the y-axes for Figure 2 and provided more descriptive captions for the figures.
>
> **References**:
>
> [1] Lanctot, Marc, et al. "A unified game-theoretic approach to multiagent reinforcement learning."
>
> [2] Sokota, Samuel, et al. "A Unified Approach to Reinforcement Learning, Quantal Response Equilibria, and Two-Player Zero-Sum Games."
>
> [3] Heinrich, Johannes, and David Silver. "Deep reinforcement learning from self-play in imperfect-information games."
>
> [4] Chen, Jingxiao, et al. "Offline Fictitious Self-Play for Competitive Games."
>
> Thanks again for your comments. We are looking forward to having further discussions with you.

---

> > ### Comment · Reviewer_CuY7 · 2025-04-09
> >
> > Thanks for the clarifications and additional results. Both of my concerns are somewhat mitigated, and I am leaning towards Accept.

---

> > > ### Author Response · Authors · 2025-04-09
> > >
> > > We are delighted that our response and additional results help to mitigate your concerns. Thank you again for providing the valuable comments and helping us further improve this paper.

---

### Official Review · Reviewer_UYuk · 2025-03-14

**Overall Recommendation:** 3

**Summary:**

This paper introduces Constrained Exploitability Descent (CED), a novel model-free offline reinforcement learning algorithm for adversarial Markov games (MGs). The authors demonstrate, both theoretically and empirically, that, unlike in MDPs, an optimal policy can be learned under policy constraints in adversarial MGs. They prove that CED converges to an unexploitable min-player policy under uniform coverage without relying on generalized gradients. Experiments in multiple game scenarios validate these theoretical results, and similar to single-agent offline RL algorithms, CED can improve the behavior policy even with non-uniform data coverage.

**Claims And Evidence:**

See methods and evaluation below

**Essential References Not Discussed:**

I think the authors may already cover most of the references.

**Experimental Designs Or Analyses:**

Simple setup but sufficient to support the goal of this method

**Methods And Evaluation Criteria:**

* The proof of theorem 1 relies on the assumption that $\frac{1}{\epsilon} \rightarrow 0$, I am curious about if the empirical performance will be improved with increasing $\epsilon$.

**Other Comments Or Suggestions:**

N/A

**Other Strengths And Weaknesses:**

Strengths:
* The paper presents solid theoretical results, proving that CED converges to a stationary point in deterministic two-player zero-sum Markov games, given the assumption of uniform data coverage.

* CED does not rely on generalized gradient computation.

Weaknesses:
* Theorem 5.2 only provides asymptotic convergence analysis under assumption of uniform coverage

**Questions For Authors:**

* Could you elaborate more on the novelty of the proposed method? It seems that the contribution is in terns of the improvement.

**Relation To Broader Scientific Literature:**

Contribute to game theory/ multi-agent in offline RL

**Theoretical Claims:**

Seems correct to me.

---

> ### Author Rebuttal · Authors · 2025-04-01
>
> Thank you for reviewing this paper and providing valuable comments. We are updating the manuscript according to the comments from all reviewers. Here, we reply to your question and concern about the paper.
>
> **[Questions For Authors]**
>
> _**Question**_: Could you elaborate more on the novelty of the proposed method? It seems that the contribution is in terms of the improvement.
>
> Yes, we can provide more explanations on the novelty of CED. While the proposed method resembles ED, the learning behaviors are quite different. ED exhibits best-iterate convergence, while CED guarantees last-iterate convergence. The improvement from best-iterate convergence to last-iterate convergence is usually at the sacrifice of policy optimality. Actually, after we apply divergence regularization to the computation of the min-player policy $\nu$, the convergence of the max-player policy $\mu$ improves, but the convergent point is no longer a Nash equilibrium policy. The CED method, however, is established upon the surprising observation that the opponent policy $\nu$ can instead preserve the property of being unexploitable as long as no explicit regularization is applied to the update of $\mu$. That is why we apply direct policy constraint on $\mu$ rather than policy penalty under the offline setting. The proposed method eventually guarantees last-iterate convergence, policy unexploitability, and bounded distributional shift at the same time.
>
> **[Methods And Evaluation Criteria]**
>
> _**Comment**_: The proof of theorem 1 relies on the assumption that $\frac{1}{\epsilon}\to\infty$, I am curious about if the empirical performance will be improved with increasing $\epsilon$.
>
> Thank you for this comment. Actually, while the assumption $\frac{1}{\epsilon}\to\infty$ theoretically guarantees the convergence of CED, it is not a necessary condition in practical games. As is stated in the second paragraph of Section 6.3, it does not affect convergence to use a small (but not too small) regularization parameter $\epsilon$ in the soccer game. As is shown in Figure 3 (right), while CED cannot converge under $\epsilon=10^{-4}$, it actually converges when $\epsilon\geq 10^{-3}$. Besides, the performance under $\epsilon=10^{-3}$ is better than under $\epsilon=10^{-2}$. Therefore, the empirical performance of CED is not guaranteed to improve with increasing $\epsilon$.
>
> Empirically, there is a trade-off with respect to the selection of $\epsilon$. If $\epsilon$ is very large, the computation of $\nu$ is reduced to behavior cloning. The equilibrium property of the converged $\bar{\nu}$ is no longer guaranteed since the interior-point premise for Theorem 5.6 can hardly be satisfied. From our experience, as long as $\epsilon$ is sufficiently large for the practical convergence of CED, a relatively small $\epsilon$ generally guarantees a relatively low NashConv of the learned policy $\nu$ under the same number of iterations.
>
> Thanks again for your comments. We are looking forward to having further discussions with you.

---

### Decision · Program_Chairs · 2025-05-01

**Decision:**

Accept (poster)

**Comment:**

The paper introduces Constrained Exploitability Descent, a model-free reinforcement learning algorithm that exhibits last iterate convergence for two player stochastic games. The authors demonstrate, both theoretically and empirically, that, an optimal policy can be learned using their method. All reviewers were positive or slightly positive about this work, so we recommend acceptance.